# Medium- and Long-Term Effects of Dapagliflozin on Serum Uric Acid Level in Patients with Type 2 Diabetes: A Real-World Study

**DOI:** 10.3390/jpm13010021

**Published:** 2022-12-22

**Authors:** Shihan Wang, Tao Yuan, Shuoning Song, Yanbei Duo, Tianyi Zhao, Junxiang Gao, Yong Fu, Yingyue Dong, Weigang Zhao

**Affiliations:** Department of Endocrinology, Key Laboratory of Endocrinology of Ministry of Health, Peking Union Medical College Hospital, Chinese Academy of Medical Sciences and Peking Union Medical College, Beijing 100730, China

**Keywords:** serum uric acid, dapagliflozin, type 2 diabetes mellitus, sodium glucose cotransporter 2 inhibitor

## Abstract

We aimed to explore the medium- and long-term (≥12 weeks) effects of dapagliflozin on serum uric acid (SUA) level in patients with type 2 diabetes mellitus (T2DM) in the real world study and to explore the influencing factors of dapagliflozin on reducing SUA level. This observational, prospective cohort study was based on the real world. There were 77 patients included in this study. They were divided into two groups. Patients in treatment group (*n* = 38) were treated as dapagliflozin 10 mg/d combined with therapy of routine glucose-lowering drugs (GLDs), and patients in the control group (*n* = 39) were treated with their routine GLDs. All measurements of physical examinations, blood, and urine samples, including age, sex, weight, height, systolic blood pressure (SBP), diastolic blood pressure (DBP), fasting blood glucose (FBG), glycosylated hemoglobin (HbA1c), and SUA, were collected at baseline for all patients in these two groups and repeated after 12, 24, and 48 weeks of therapy. We compared the changes of metabolic indicators including SUA in these two groups to evaluate the effects of dapagliflozin and analyzed its influencing factors. In the dapagliflozin group, mean SUA levels significantly decreased from 334.2 ± 99.1 μmol/L at baseline to 301.9 ± 73.2 μmol/L after 12 weeks therapy (t = 2.378, *p* = 0.023). There was no significant statistical difference of SUA levels after 24 weeks treatment of dapagliflozin compared with 12-week and 48-week treatment with dapagliflozin (*p* > 0.05). We found that baseline SUA had a significant impact on the effect of dapagliflozin on reducing SUA (OR 1.014, 95%CI 1.003–1.025, *p* = 0.014) by logistic regression analysis. Receiver operating characteristic (ROC) curve showed that T2DM patients with SUA level ≥ 314.5 μmol/L had relative accuracy in recognizing the good effects of dapagliflozin on reducing SUA (sensitivity 76.9%, specificity 76.2%). Combination therapy of dapagliflozin with routine blood-glucose-lowering drugs in T2DM patients showed the significant and sustained stable effect of lowering SUA level in this real-world study.

## 1. Introduction

Diabetes mellitus (DM) is a complex metabolic syndrome characterized by high blood sugar levels. In 2021, the prevalence of DM was 10.5%. It is predicted that the prevalence in 2045 may reach 12.2%, that is, about 783.2 million people [1]. Moreover, DM accounted for 9.9% of all-cause mortality among people aged 20 to 99 years globally [2], which placed huge burdens on the global health system. Studies found that SUA level was a direct risk factor for T2DM, which was independent of other established risk factors such as diet and metabolic syndrome (MetS) [3]. Under normal purine diet, the normal SUA level of adults was generally less than 420 μ mol/L (7 mg/dL). Meta-analysis demonstrated that the risk of T2DM increased by 6% per 59.5 μmol/L increment in SUA level (RR 1.06, 95% CI 1.04–1.07) by dose–response analysis [4]. In addition, the high level of SUA was closely related to cardiovascular and cerebrovascular diseases, MetS, diabetes nephropathy (DN), diabetes microvascular disease, and diabetes peripheral neuropathy [5,6,7,8]. Meta-analysis showed that the risk of vascular complications increased by 28% (OR 1.28, 95%CI 1.12–1.46), and mortality increased by 9% (HR 1.09, 95%CI 1.03–1.16) in DM patients per 100 μmol/L increment in SUA level [9], and all these indicated risks were independent of normal basal SUA levels. Therefore, actively controlling the level of SUA in patients with T2DM was helpful to reduce DM and its complications and avoid worsening of metabolic disorder.

For patients with T2DM, we advocated precision medicine. The ideal GLDs can not only decrease blood glucose but also reduce the occurrence of other metabolic disorders and related complications. Sodium-glucose cotransporter-2 inhibitor (SGLT-2i) has attracted much attention as a new type of GLD. It is unique because of its independence of insulin secretion and action, working by inhibiting the reabsorption of glucose in proximal renal tubules of the kidney. SGLT-2i has shown good effects of losing weight, reducing blood pressure, resisting inflammation, improving insulin resistance, and protecting the cardiovascular system and kidney [10,11]. Although drugs such as febuxostat and allopurinol can reduce SUA, they do not decrease the incidence of cardiovascular and cerebrovascular events, and febuxostat may even increase these events [12]. At present, some studies have pointed out that SGLT-2i has a strong potential to reduce the level of SUA, but as far as we know, they were all clinical, well-controlled trials and meta analyses, and there was seldom any real-world study.

In this study, we aimed to explore the medium- and long-term (≥12 weeks) effects of SGLT-2i (dapagliflozin) on SUA level in patients with T2DM in the real world and explored the influencing factors of dapagliflozin on reducing SUA level.

## 2. Materials and Methods

This observational, prospective cohort study was conducted in the Peking Union Medical College Hospital (PUMCH) population from January 2021 to July 2022. These subjects were recruited from individuals aged at least 18 years old, who were definitely diagnosed with T2DM, and whose estimated glomerular filtration rate (eGFR) was ≥ 60 mL/min/1.73 m^2^ without contraindications to SGLT2i. Patients were educated to avoid all beverages and foods with high fructose and purine in the routine clinic follow-up as much as possible.

Exclusion criteria were as follows: other types of DM; acute complications of T2DM within the last 6 months; those were receiving ongoing treatment with SGLT2 inhibitors and uric-acid-lowering drugs (including allopurinol, febuxostat and benzbromarone); severe liver and kidney dysfunction; malignant tumors and other serious diseases; and pregnant or lactating women.

There were 77 subjects included in this study. They were divided into two groups. The baseline HbA1c of most patients with many GLDshad not yet reached the standard, and the use of dapagliflozin could became a treatment option. In addition, based on the guideline of type 2 diabetes mellitus therapy in clinical practice, patients with cardiovascular and cerebrovascular complications or high risks of atherosclerosis cardiovascular or kidney disease were preferentially included in the dapagliflozin group. Patients in the treatment group (*n* = 38) were treated with dapagliflozin 10 mg/d added on the basis of routine GLDs, and patients in the control group (*n* = 39) continued their routine GLDs. These GLDs were metformin, sulphonylureas, glinides, glitazones, α-glucosidase inhibitors, dipeptidyl peptidase 4 inhibitor (DPP-4i), glucagon-like peptide-1 receptor agonist (GLP-1RA), and insulin. Dosages of the drugs were adjusted according to fasting blood glucose and random blood glucose level testing by self-monitor of blood glucose.

All measurements of physical examinations and blood and urine samples were collected at baseline in all subjects in these two groups and repeated after 12, 24, and 48 weeks of treatment. The study flow chart is shown in Figure 1. Medical records were accessed for baseline and each clinic follow-up, including weight, height, SBP, and DBP. Parameters such as FBG, HbA1c, and SUA levels were also recorded. Body mass index (BMI) was calculated as weight (kg) divided by the square of the height in meters (m^2^).

All clinical and laboratory parameters were determined by using standard guidelines and methods by PUMCH. This study was approved by the PUMCH Ethics Committee (Number: S-2574). All participants signed written informed consent voluntarily.

## 3. Statistical Analyses

For continuous variables, Kolmogorov–Smirnov Z-test was conducted. Once normal distribution was satisfied, values were expressed as the mean ± standard deviation. Otherwise, values were expressed as the median (interquartile range). Categorical variables were expressed in *n* (%). To compare the characteristics in the two groups of the cohort, Mann–Whitney U-test was used for non-parametric variables. Student’s *t*-test was used for parametric variables, and the chi-squares test was used for categorical variables. One-way repeated measures ANOVA and paired student’s *t*-test were used to explore the within-group differences in SUA levels, weight, SBP, DBP, and HbA1c before and after dapagliflozin therapy and routine GLDs therapy. We further used logistic regression analysis to evaluate the factors Influencing the efficacy of dapagliflozin on reducing SUA. A *p*-value less than 0.05 was considered statistically significant. All statistical analyses were carried out by using the statistical program SPSS (version 26, SPSS, Chicago, IL, USA).

## 4. Results

Of the 77 participants who enrolled in the study, 38 patients received daily dapagliflozin 10 mg therapy on the basis of routine GLDs. The demographic and baseline characteristics, including age, sex, BMI, diabetes duration, FBG, HbA1c, serum creatinine, SBP, DBP, urine ratio of albumin/creatinine, incidence of hypertension, incidence of microvascular complications, and application of GLDs background, in the treatment group were comparable to 39 patients who were treated with their routine GLDs (*p* > 0.05) in the control group. Because patients with cardiovascular and cerebrovascular complications or high risks of atherosclerosis cardiovascular or kidney disease were preferentially included in the dapagliflozin group, the incidence of macrovascular complications of dapagliflozin group was higher than the routine GLDs control group (65.8% vs. 25.6%; χ^2^ = 12.513, *p* < 0.001). The total cholesterol (4.6 ± 1.0 vs. 4.1 ± 1.0 mmol/L; t = 2.197, *p* = 0.031) and low-density lipoprotein cholesterol (LDL-C) (2.7 ± 0.8 vs. 2.2 ± 0.8 mmol/L; t = 2.859, *p* = 0.006) of routine GLDs group were significantly higher than dapagliflozin group. Baseline SUA levels were similar between the two groups (334.2 ± 99.1 vs. 352.8 ± 70.0 μmol/L; t = −0.929, *p* = 0.356) (Table 1).

Thirty-seven patients (97.4%) in the dapagliflozin group achieved 12 weeks of medication and were followed-up as planned, and the routine GLDs group was followed-up at 12 weeks. Twenty-three patients (60.5%) in the dapagliflozin group and twenty-two patients (56.4%) in the control group achieved 24 weeks of medication and were followed-up. Fourteen patients (36.8%) in the dapagliflozin group and thirteen patients (33.3%) in the control group achieved 48 weeks of medication and were followed-up. The number of completed follow-up cases was similar between the two groups and comparable. In the dapagliflozin group, mean SUA level decreased from 334.2 ± 99.1 to 301.9 ± 73.2 μmol/L (t = 2.378, *p* = 0.023), and in the control group, mean SUA level changed from 352.8 ± 70.0 to 353.9 ± 64.1 μmol/L (t = −0.272, *p* = 0.787) after 12 weeks of therapy. Mean SUA level reduction from baseline was significantly greater for the dapagliflozin group than the GLDs group (t = −3.300, *p* = 0.001), while weight, FBG, HbA1c, serum creatinine, SBP, DBP, and urine ratio of albumin/creatinine did not change much in these two groups after 12 weeks of therapy (*p* > 0.05). In the treatment group, weight (74.1 ± 12.1 vs. 72.4 ± 11.7 kg; t = 4.305, *p* < 0.001), SBP (135.7 ± 20.3 vs. 129.9 ± 15.8 mmHg; t = 2.242, *p* = 0.032), DBP (83.7 ± 12.7 vs. 79.4 ± 12.3 mmHg; t = 2.330, *p* = 0.026), and HbA1c (8.0 ± 1.0 vs. 6.9 ± 1.0%; t = 5.619, *p* < 0.001) after 12 weeks treatment of dapagliflozin were lower than the baseline. Meanwhile, these demographic and general laboratory blood parameters such as SUA level, weight, SBP, and DBP after 24 weeks treatment of dapagliflozin had no significant statistical differences compared with 12 weeks treatment of dapagliflozin and 48 weeks treatment of dapagliflozin (*p* > 0.05). In the control group, HbA1c after 12-week treatment of GLDs was lower than the baseline (7.7 ± 1.6 vs. 6.6 ± 0.7%; t = 4.482, *p* < 0.001), while parameters such as SUA level, weight, SBP, and DBP at baseline had no significant statistical differences compared with 12, 24, and 48 weeks of treatment with routine GLDs (*p* > 0.05) (Table 2) (Figure 2).

Taking SUA decreased by 10% from baseline to 12 weeks of treatment as the threshold for the sub-group of patients whose SUA levels decreased by 10% or more, and their baseline SUA (381.8 ± 94.7 vs. 293.8 ± 73.0 μmol/L; t = 3.047, *p* = 0.005) was higher significantly, and their baseline HbA1c (7.4 ± 0.9 vs. 8.3 ± 1.0%; t = −2.610, *p* = 0.014) was lower than the sub-group whose SUA levels decreased less than 10%. In addition, the SUA decreased by 10% or more sub-group had higher weight (79.1 ± 13.7 vs. 70.8 ± 9.6 kg; t = 2.037, *p* = 0.05) and higher incidence of hypertension (92.3% vs. 57.1%, *p* = 0.051) compared to the SUA decreased less than 10% sub-group, but without statistically difference (Table 3).

Logistic regression analysis (including body weight, HbA1c, SUA, and hypertension incidence, using the forward LR method) found that baseline uric acid had a significant impact on the effects of dapagliflozin on reducing SUA (OR 1.014, 95% CI 1.003–1.025, *p* = 0.014).

The ROC curve of the baseline SUA was drawn according to the decreased levels of serum uric acid ≥ 10% after 12 weeks of dapagliflozin, and we found that baseline SUA ≥ 314.5 μmol/L had a good accuracy in identifying T2DM patients who may have experienced uric-acid-lowering effects after dapagliflozin therapy, with a sensitivity of 76.9% and a specificity of 76.2% (Figure 3).

## 5. Discussion

SGLT-2i were widely used as monotherapy and combination therapy in patients with T2DM. In our study, dapagliflozin worked as a combination therapy added to other GLDs, and HbA1c decreased from 8.0 ± 1.0% at baseline to 6.9 ± 1.0% at 12 weeks of therapy (t = 5.619, *p* < 0.001), which showed its excellent glucose-lowing effect. In terms of side effects, posthitis was present in one male patient (2.6%) after 2-month application of dapagliflozin. After active anti-infection treatment, his symptoms completely improved, and he resumed dapagliflozin. Side effects such as hypoglycemia and hypotension were not observed in the subjects in the dapagliflozin group.

Various clinical studies showed dapagliflozin had a strong potential to reduce the level of SUA. Our previous study aimed to evaluate the effects of 1 week of therapy with dapagliflozin (dose: 10 mg daily) on SUA and found that SUA decreased from 347.75 ± 7.75 μmol/L at baseline to 273.25 ± 43.18 μmol/L at the end of therapy (*p* = 0.001), and urine fraction excretion of uric acid (FEUA) increased from 0.009 to 0.029 (*p* = 0.035) [13]. Mazhar Hussain explored the urate-lowering effect after applying dapagliflozin (dose: 5 to 10 mg daily) or empagliflozin for 4 weeks and found that SUA went from 422.45 ± 107.1 μmol/L to 404.6 ± 130.9 (*p* = 0.001); however, this trial design considered dapagliflozin and empagliflozin as one group, so it was not possible to individually assess the uric-acid-lowering effect of dapagliflozin [14]. Serge A Jabbour showed that therapy with dapagliflozin (dose: 10 mg daily) led to a reduction in SUA of 44.63 μmol/L at 24 weeks and 45.22 μmol/L at 48 weeks [15]. Another article found that the reduction in uric acid was 26.18 μmol/L at 24 weeks (dose: 10 mg daily) [16]. Clifford J Bailey found that the SUA after 102 weeks of treatment of dapagliflozin (dose: 10 mg daily) decreased by 52.9 μmol/L [17]. Some meta-analyses also support the uric-acid-lowering effect of dapagliflozin. A meta-analysis of 55 studies found that the weighted mean difference (WMD) of dapagliflozin for lowering SUA was 35.17 μmol/L (95% CI 30.66–39.68) [18] and an article reviewing 62 studies showed that the WMD was 36.99 μmol/L (95% CI 32.25–41.73) [19]. Yakai Xin integrated 31 studies and found that the WMD was 38.05 μmol/L (95% CI 31.62–44.47) [20]. Those studies were almost well-controlled randomized clinical trials.

Our research was based on the real world, which was helpful in exploring the effects of dapagliflozin on reducing SUA in the real treatment environment. In our previous study, we found that the lowering effect of dapagliflozin on SUA appeared quickly after only one week of therapy [13]. In this real-word study, we found that the benefit of lowering SUA of dapagliflozin, with an average dose of 23.1 μmol/L at 12 weeks, persisted until 48 weeks of follow-up. From our study, it seem that dapagliflozin showed its benefits of lowing SUA within a short treatment time, and this benefit could be maintained. In addition, we also found long-term benefits of dapagliflozin in reducing blood pressure and weight.

Some studies found that the key factors influencing the effect of SGLT2i in reducing SUA were as follows [19]: (1) The longer the duration of T2DM, the worse the effect of reducing SUA; (2) the higher the HbA1c, the worse the effect of reducing SUA; (3) the lower the eGFR, the worse the effect of reducing SUA; and (4) dapagliflozin showed a special dose-dependent effect; that is, with the increment of the applied dose, the effect of reducing SUA was also enhanced. Our study found that the effect of dapagliflozin in reducing SUA may be related to baseline HbA1c, SUA level, body weight, and hypertension complications. Logistic regression analysis showed that baseline SUA had a significant impact on the effect of dapagliflozin on reducing SUA (OR 1.014, 95% CI 1.003–1.025, *p* = 0.014). ROC curve showed that T2DM combined with SUA level ≥ 314.5 μmol/L has good accuracy in recognizing the good effect of dapagliflozin on reducing SUA (sensitivity 76.9%; specificity 76.2%). This may suggest sufficient benefits of dapagliflozin when used in T2DM patients with high basal SUA levels in clinical real-world practice.

The current assumption of mechanisms of dapagliflozin in lowering SUA levels were as follows: (1) In the mouse model, tubular perfusion of D-glucose reduced the fractional urate absorption in the proximal convoluted tubule [21]. In clinical research, scientists also found that the increase of urine sugar caused the increase of uric acid excretion, which was considered to be mediated by glucose transporter 9 isoform 2 (GLUT9b) and urate transporter 1 (URAT1) or other transporters in the proximal tubule, and inhibits the reabsorption of uric acid mediated by GLUT9b in the collecting duct [22,23,24]. (2) The level of SUA in diabetes patients was higher than that in non-diabetes patients, which may be related to the excessive production of uric acid caused by the high activity of xanthine oxidase and lipid peroxidation and decreased excretion of uric acid caused by kidney damage related to diabetes. Dapagliflozin can improve insulin resistance, weight loss, kidney protection, and other benefits, thereby reducing SUA. In our study, at 12 weeks, we could observe the benefits of weight loss and pressure reduction, with an average weight loss of 2.74 kg, SDP of 7.7 mmHg, and DBP of 5.09 mmHg, and these benefits persisted after 48 weeks of follow-up.

Some studies showed that the lowered level of SUA had benefits in patients with T2DM. A randomized, placebo-controlled study from Hong Kong found that lowering the levels of SUA with allopurinol reduced the rate of eGFR loss [25]. Further, Yan Miao suggested that the risk of renal events (a doubling of serum creatinine or end-stage renal disease) was decreased by 6% (95% CI 3–10%) per 30 μmol/L decrease in SUA level in T2DM and nephropathy patients [26].

Our study was limited in that we did not check urine for a complete examination for uric acid, and a relatively small sample size limited the generalization of the study, so the conclusions needed to be carefully interpreted. Moreover, we did not further explore whether there were decreases in the complication risks associated with lowering SUA in T2DM patients in the real world, which could necessitate further and more studies.

## 6. Conclusions

In this real-world study, we found that dapagliflozin, as a combination therapy of type 2 diabetes, can reduce the level of SUA, and this benefit can be stable for a long time. We suggest that dapagliflozin can be used in combination therapy with T2DM patients with high SUA levels.

## Figures and Tables

**Figure 1 jpm-13-00021-f001:**
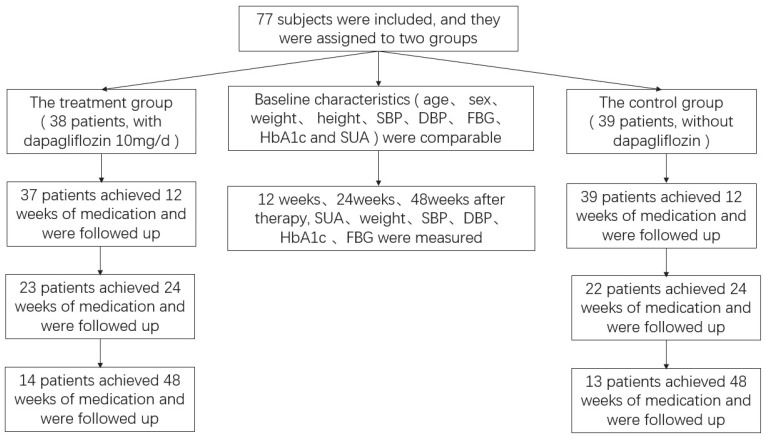
Study flow chart. Abbreviations: SBP, systolic blood pressure; DBP, diastolic blood pressure; HbA1C, glycosylated hemoglobin; FBG, fasting blood glucose; SUA, serum uric acid.

**Figure 2 jpm-13-00021-f002:**
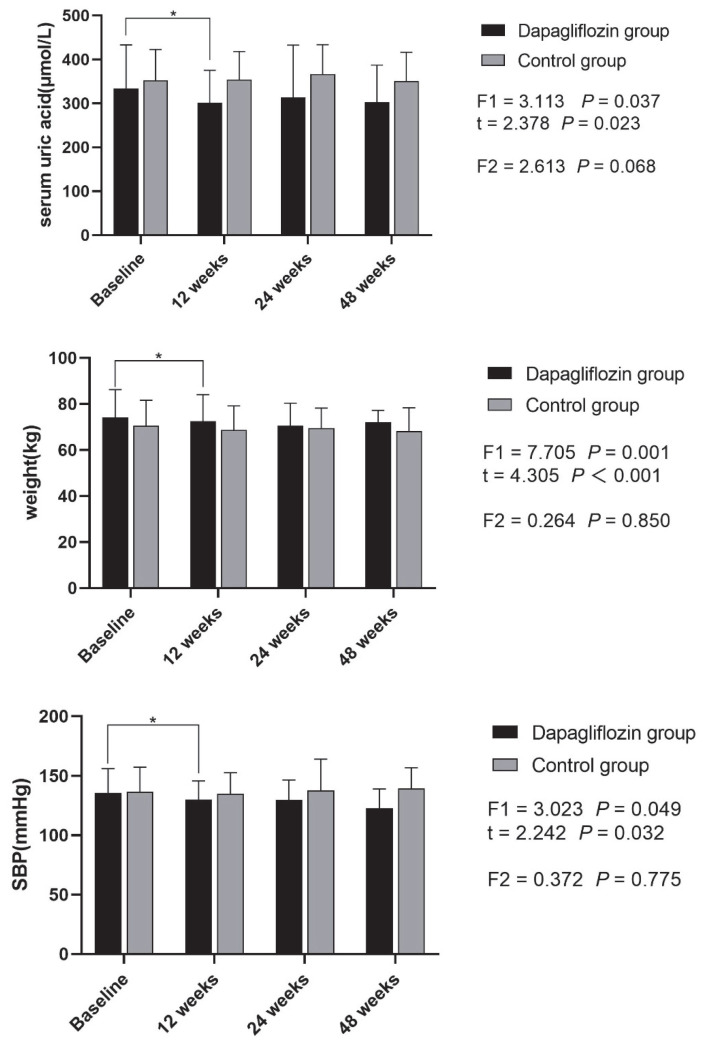
Changes of serum uric acid, weight, SBP, DBP, and HbA1c in dapagliflozin group and control group before and after treatment. F1, one-way repeated measures ANOVA in dapagliflozin group; F2, one-way repeated measures ANOVA in control group. * *p* < 0.05.

**Figure 3 jpm-13-00021-f003:**
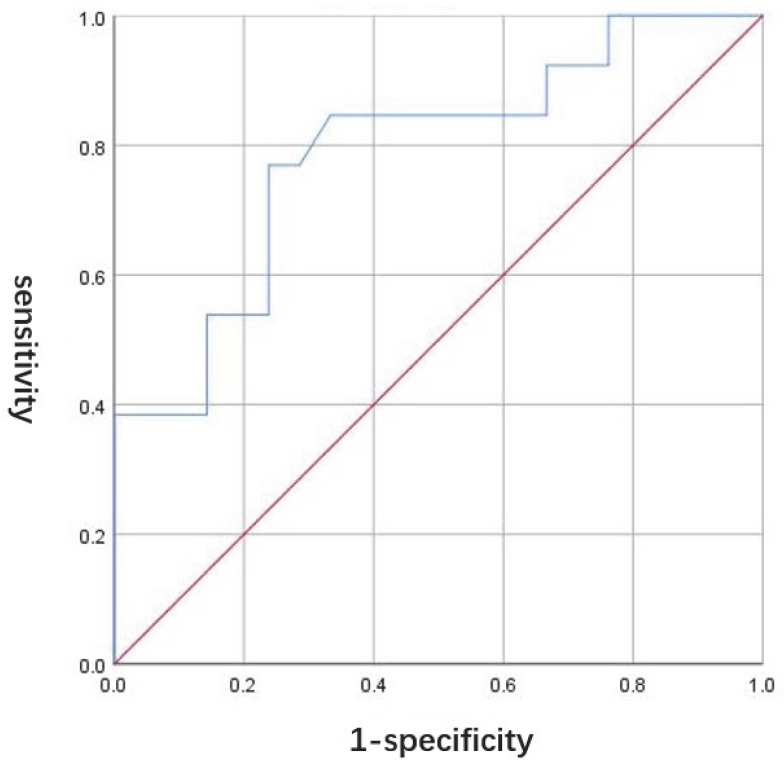
The ROC curve of the baseline SUA and the decreased levels of serum uric acid ≥ 10% after 12 weeks of dapagliflozin. Red line: reference line; blue line: ROC curve of 12-week therapy of dapagliflozin.

**Table 1 jpm-13-00021-t001:** Demographic and baseline characteristics in the dapagliflozin group (Dapagliflozin + GLDs) and control group (GLDs).

Variable	Dapagliflozin + GLDs (*n* = 38)	GLDs (*n* = 39)	Statistic	*p*-Value
Age (years)	57.5 (46.3, 66.0)	57.6 ± 13.3	0.245	0.807
Gender (males, %)	20 (52.6%)	16 (41.0%)	0.630	0.427
Diabetes duration (years)	3 (2, 4)	2 (2,4)	−2.086	0.057
Height (cm)	168.0 ± 8.2	168.0 (157.0, 172.0)	−1.380	0.168
Weight(kg)	74.1 ± 12.1	70.5 ± 11.2	1.352	0.180
BMI (kg/m^2^)	26.3 ± 4.0	25.6 (23.4, 27.1)	−0.408	0.683
Total cholesterol (mmol/L)	4.1 ± 1.0	4.6 ± 1.0	−2.197	0.031
Triglycerides(mmol/L)	1.3 (0.9, 2.0)	1.7 ± 0.8	0.800	0.424
LDL-C (mmol/L)	2.2 ± 0.8	2.7 ± 0.8	−2.859	0.006
HDL-C (mmol/L)	1.1 ± 0.2	1.2 (0.9, 1.3)	0.397	0.691
Fasting blood glucose (mmol/L)	8.4 (7.3,11.2)	7.8 (7.0, 9.8)	−1.562	0.118
HbA1c (%)	8.0 ± 1.0	7.7 ± 1.6	1.036	0.304
Serum creatinine (μmol/L)	62.5 ± 11.7	62.0 ± 10.3	0.170	0.886
Serum uric acid (μmol/L)	334.2 ± 99.1	352.8 ± 70.0	−0.929	0.356
Urine ratio of albumin/creatinine	8.0 (4.0, 34.0)	9.0 (5.0, 16.0)	−0.145	0.885
Hypertension (*n*, %)	27 (71.1%)	21 (53.8%)	2.427	0.119
Systolic blood pressure (mmHg)	135.7 ± 20.3	136.8 ± 20.9	−0.237	0.813
Diastolic blood pressure (mmHg)	83.7 ± 12.7	82.9 ± 10.6	0.284	0.777
Macrovascular complications (*n*, %)	25 (65.8%)	10 (25.6%)	12.513	<0.001
Microvascular complications (*n*, %)	5 (13.2%)	0	3.535	0.06
Background GLD (*n*, %)				
Metformin	32 (84.2)	33 (84.6%)	0.002	0.961
Sulphonylureas	8 (21.1%)	10 (25.6%)	0.226	0.634
Glinides	1 (2.6%)	3 (7.69%)	0.237	0.626
Glitazones	1 (2.6%)	1 (2.6%)	-	1.00
α-glucosidase inhibitor	20 (52.6%)	14 (35.9%)	2.186	0.139
DPP-4i	10 (26.3%)	7 (17.9%)	0.783	0.376
GLP-1RA	3 (7.9%)	0	1.442	0.230
Insulin	14 (36.8%)	11 (28.2%)	0.655	0.418

Abbreviations: GLDs, glucose-lowering drugs; BMI, body mass index; LDL-C, low-density lipoprotein cholesterol; HDL-C, high-density lipoprotein cholesterol; HbA1c, glycosylated hemoglobin; DPP-4i, dipeptidyl peptidase 4 inhibitor; GLP-1RA, glucagon-like peptide-1 receptor agonist.

**Table 2 jpm-13-00021-t002:** Comparison of parameters in the dapagliflozin group (Dapagliflozin + GLDs) and control group (GLDs) before and after therapy.

Variable	Baseline Dapagliflozin (*n* = 38) Control (*n* = 39)	12 Weeks Dapagliflozin (*n* = 37) Control (*n* = 39)	24 Weeks Dapagliflozin (*n* = 23) Control (*n* = 22)	48 Weeks Dapagliflozin (*n* = 14) Control (*n* = 13)
Uric acid (μmol/L)				
Dapagliflozin group	334.2 ± 99.1	301.9 ± 73.2 ^a^	313.3 ± 119.4	295.5 ± 84.8
Control group	352.8 ± 70.0	353.9 ± 64.1	367.1 ± 66.5	351.2 ± 65.2
Uric acid decreased compared with the previous follow-up (μmol/L) 95% CI				
Dapagliflozin group		23.1 (3.4, 42.7)	−9.3 (−36.1, 17.5)	33.5 (−0.16, 67.2)
Control group		2.6 (−17.1, 22.4)	4.2 (−16.0, 24.3)	31.4 (−7.1, 70.0)
Weight (kg)				
Dapagliflozin group	74.1 ± 12.1	72.4 ± 11.7 ^a^	70.6 ± 9.7	72.0 ± 5.2
Control group	70.5 ± 11.2	68.7 ± 10.4	69.6 ± 8.6	68.2 ± 10.1
Systolic blood pressure (mmHg)				
Dapagliflozin group	135.7 ± 20.3	129.9 ± 15.8 ^a^	129.8 ± 16.8	122.8 ± 16.2
Control group	136.8 ± 20.9	135.1 ± 17.5	137.6 ± 26.5	139.3 ± 17.7
Diastolic blood pressure (mmHg)				
Dapagliflozin group	83.7 ± 12.7	79.4 ± 12.3 ^a^	80.0 ± 11.5	79.1 ± 9.4
Control group	82.9 ± 10.6	82.6 ± 9.5	81.9 ± 10.8	85.6 ± 7.7
HbA1C (%)				
Dapagliflozin group	8.0 ± 1.0	6.9 ± 1.0 ^a^	6.7 ± 0.9	6.6 ± 0.6
Control group	7.7 ± 1.6	6.6 ± 0.7 ^a^	6.4 ± 0.8	6.3 ± 0.6

^a^ Compared with baseline; *p* < 0.05.

**Table 3 jpm-13-00021-t003:** Factors influencing the efficacy of dapagliflozin in reducing serum uric acid.

Variable	Uric Acid Decreased ≥ 10% (*n* = 13)	Uric Acid Decreased < 10% (*n* = 21)	Statistic	*p*-Value
Age (years)	53.0 ± 10.5	59.3 ± 10.7	−1.686	0.102
Gender (male, %)	6 (46.2%)	12 (57.1%)	-	0.725
Diabetes duration (years)	3.0 (2.0, 4.0)	4.0 (2.5, 4.0)	0.400	0.420
Height (cm)	168.9 ± 9.1	167.5 ± 8.2	0.432	0.669
Weight(kg)	79.1 ± 13.7	70.8 ± 9.6	2.037	0.050
BMI (kg/m^2^)	27.6 ± 3.2	25.4 ± 4.4	1.528	0.137
Total cholesterol (mmol/L)	4.3 ± 1.0	3.9 ± 0.8	1.443	0.159
Triglycerides (mmol/L)	1.8 (1.0, 2.8)	1.2 (0.8, 1.8)	0.238	0.242
LDL-C (mmol/L)	2.3 ± 1.0	2.1 ± 0.7	0.811	0.424
HDL-C (mmol/L)	1.0 ± 0.2	1.1 ± 0.3	−1.224	0.230
Fasting blood glucose (mmol/L)	8.7 ± 1.8	9.7 ± 3.1	−1.021	0.315
HbA1c (%)	7.4 ± 0.9	8.3 ± 1.0	−2.610	0.014
Serum creatinine (μmol/L)	66.4 ± 13.4	60.8 ± 11.4	1.266	0.215
Uric acid (μmol/L)	381.8 ± 94.7	293.8 ± 73.0	3.047	0.005
Urine ratio of albumin/creatinine	7.5 (3.8, 12.0)	8.0 (4.0, 37.3)	0.531	0.555
Hypertension (*n*, %)	12 (92.3%)	12 (57.1%)	-	0.051
Systolic blood pressure (mmHg)	139.7 ± 19.3	135.0 ± 21.7	0.663	0.513
Diastolic blood pressure (mmHg)	88.6 ± 12.2	80.0 ± 12.7	1.987	0.057

Abbreviations: BMI, body mass index; LDL-C, low-density lipoprotein cholesterol; HDL-C, high-density lipoprotein cholesterol; HbA1c, glycosylated hemoglobin.

## Data Availability

Data available on request due to restrictions.

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
