# Peer review of "Medium- and Long-Term Effects of Dapagliflozin on Serum Uric Acid Level in Patients with Type 2 Diabetes: A Real-World Study"

_jpm, 2022, doi:10.3390/jpm13010021_

Round 1

Reviewer 1 Report

Dear authors,

Thank you for your interesting study concerning SGLT2i and levels of SUA in type 2 diabetes patients.

The topic is of interest, the study well done and described.

I have only a few comments.

Firstly, concerning form:

-Line 136: some numbers are written in letters and some in numbers. It would be better to choose letters or numbers and write it the same way throughout the article.

-Line 139: a capital letter is missing

Then, concerning the study itself:

- I have not seen the ROC curve in the article. Maybe it would be interesting to add it.

-In the discussion, maybe you could explain better why lowering SUA could be important in diabetic patients. You say several time in the text your study is a real-life study, which is actually interesting, but then what could be the benefit in the real life for patients to have a SUA lowered by treatment? You explain in the introduction that high levels of SUA are associated with diabetes complications, but we don't know if some studies have been done concerning lowering the levels of SUA and studying the implication in diabetes complications.

Author Response

Dear reviewers:

    On behalf of my co-authors, we appreciate reviewers very much for constructive comments and suggestions on our manuscript entitled “Medium and long-term effects of dapagliflozin on serum uric acid level in patients with type 2 diabetes: A Real-World Study” (Manuscript ID: jpm-2115505).

    We have studied the comments carefully and made corrections which we hope meet with approval. The corrections are in the manuscript and responses to the reviewers’ comments are as follows.

Replies to reviewers’ comments:

Reviewer #1

1.-Line 136: some numbers are written in letters and some in numbers. It would be better to choose letters or numbers and write it the same way throughout the article.

-Line 139: a capital letter is missing

Response: Thanks for your careful checks. We are sorry for our carelessness. Based on your comments, we have made corrections to make the numbers harmonized within the whole manuscript.

  1. - I have not seen the ROC curve in the article. Maybe it would be interesting to add it.

Response: As suggested by the reviewer, we have added the ROC curve.

  1. -In the discussion, maybe you could explain better why lowering SUA could be important in diabetic patients. You say several time in the text your study is a real-life study, which is actually interesting, but then what could be the benefit in the real life for patients to have a SUA lowered by treatment? You explain in the introduction that high levels of SUA are associated with diabetes complications, but we don't know if some studies have been done concerning lowering the levels of SUA and studying the implication in diabetes complications.

Response: We sincerely appreciate the valuable comments. We did not further explore whether there were decreases in the risks associated with uric acid, which is one of the limitations of our manuscript. We recognize this limitation should be mentioned in the paper, so we added the following sentences “Also, We did not further explore whether there were decreases in the complication risks associated with lowing SUA in T2DM patients in real world , which could need further and more studies” at line 283-285. If possible, we will focus on this question in the following research.                                                  

We have added more reference on studies concerning lowering the levels of SUA and the implication in diabetes nephropathy into the discussion part in the revised manuscript. Line 275-280, Some studies showed that the lowing level of SUA had benefits in patients with T2DM. A randomized, placebo-controlled study from Hong-Kong found that lowering the levels of SUA by allopurinol reduced the rate of eGFR loss. AND Yan Miao suggested that the risk of renal events (a doubling of serum creatinine or end-stage re-nal disease) was decreased by 6%95%CI 3-10%per 30μmol/L decrement in SUA level in T2DM and nephropathy patients were added.

Reviewer 2 Report

 12 – 15 This observationalprospective cohort study was based on the real world. There were 77 patients included in this study. They were divided into two groups. Patients in treatment group (n = 38) were treated as dapagliflozin 10 mg/d combined with therapy of routine glucose-lowering drugs  (GLDs), and patients in control group (n = 39) treated as their routine GLDs.

It is not conceivable that the present study is purely observational. There must have been a basic reason for adding dapagliflozin to the conventional treatment of DM2 in the group thus treated. Even if it was an exploratory study. Readers will appreciate knowing that basic reason

39-44  Studies found that SUA level was a  direct risk factor for T2DM, which was independent of other established risk factors such as diet and metabolic syndrome (MetS)[3]. Meta-analysis demonstrated that the risk of  T2DM increased by 6% per 59.5μmol/L (1 mg/dl) increment in SUA level (RR 1.06, 95% CI 42 1.04–1.07)[4]. In addition, the high level of SUA was closely related to cardiovascular and  cerebrovascular diseases, MetS, diabetes nephropathy(DN), diabetes microvascular disease and diabetes peripheral neuropathy[5-8].

Meta-analysis showed that the risk of vascular complications increased by 28%OR 1.28 95%CI 1.12-1.46and mortality increased by 9% HR 1.0995%CI1.03-1.16in DM patients per 100μmol/L increment in SUA level[9].

The range of normal uric acid levels is very wide. It is suggested to include normal reference levels and comment on whether the indicated risks are independent of normal basal uric acid levels or only exist above normal reference values.

57-59 Although drugs such as febuxostat and allopurinol can reduce SUA, they did not decrease the incidence of cardiovascular and cerebrovascular events, and even febuxostat maybe increase these events[12].

A question. If the risks of high uric acid levels are proposed as risk factors for DM2 or its complications, and uricosuric agents do not decrease its incidence, why should we assume that dapagliflozin, due to its uricosuric activity, would?

It was not very clear to me if the dapagliflozin treatment was administered for all 48 weeks of the study or only for 12 weeks.

131 – 133 Abbreviations: GLDs, glucose-lowering drugs; BMI, body mass index; LDL-C, low-density lipo-131 protein cholesterol; HDL-C, high-density lipoprotein cholesterol; HbA1c, glycosylated hemoglo-132 bin; DPP-4i, dipeptidyl peptidase 4 inhibitor; GLP-1RA, glucagon like peptide-1 receptor agonist.

It is suggested to indicate in the graphs the statistical test used and the level of significance.

Figure 2 does not indicate whether the paired tests were done after anova, for example

256 – 258 In this real-world study, we found that dapagliflozin, as a combination therapy of type 2 diabetes, can reduce the level of SUA and this benefit can be stable for a long time. We suggested that dapagliflozin can be used in combination therapy with T2DM patients with high SUA level.

It seems very clear that the best response to reduce uric acid levels is observed in subjects with high HbA1c values. In how many of the subjects studied in whom dapagliflozin was added relevant reductions were achieved that suggest a decrease in the risks associated with uric acid? How potentially beneficial could the use of dapagliflozin be in prediabetic subjects?

Author Response

Dear  reviewers:

    On behalf of my co-authors, we appreciate  reviewers very much for constructive comments and suggestions on our manuscript entitled “Medium and long-term effects of dapagliflozin on serum uric acid level in patients with type 2 diabetes: A Real-World Study” (Manuscript ID: jpm-2115505).

    We have studied the comments carefully and made corrections which we hope meet with approval. The corrections are in the manuscript and responses to the reviewers’ comments are as follows.

Replies to reviewers’ comments:

Reviewer #2

  1. -Line 12 – 15 This observational、prospective cohort study was based on the real world. There were 77 patients included in this study. They were divided into two groups. Patients in treatment group (n = 38) were treated as dapagliflozin 10 mg/d combined with therapy of routine glucose-lowering drugs (GLDs), and patients in control group (n = 39) treated as their routine GLDs.

It is not conceivable that the present study is purely observational. There must have been a basic reason for adding dapagliflozin to the conventional treatment of DM2 in the group thus treated. Even if it was an exploratory study. Readers will appreciate knowing that basic reason

Response: We think this is an excellent suggestion, we have re-written this part according to the reviewer’s suggestionline 83-85, The baseline HbA1c of most patients with many glucose-lowering drugs had not yet reached the standard, and the use of dapagliflozin could became a treatment option. In addition, based on the guideline of type 2 diabetes mellitus therapy in Clinical practice, patients with cardiovascular and cerebrovascular complications or high risks of atherosclerosis cardiovascular or kidney disease were preferentially included in the dapagliflozin group” was added.

  1. -Line 39-44 Studies found that SUA level was a direct risk factor for T2DM, which was independent of other established risk factors such as diet and metabolic syndrome (MetS)[3]. Meta-analysis demonstrated that the risk of T2DM increased by 6% per 59.5μmol/L (1 mg/dl) increment in SUA level (RR 1.06, 95% CI 42 1.04–1.07)[4]. In addition, the high level of SUA was closely related to cardiovascular and cerebrovascular diseases, MetS, diabetes nephropathy(DN), diabetes microvascular disease and diabetes peripheral neuropathy[5-8].

Meta-analysis showed that the risk of vascular complications increased by 28%(OR 1.28 95%CI 1.12-1.46)and mortality increased by 9% (HR 1.09,95%CI1.03-1.16)in DM patients per 100μmol/L increment in SUA level[9].

The range of normal uric acid levels is very wide. It is suggested to include normal reference levels and comment on whether the indicated risks are independent of normal basal uric acid levels or only exist above normal reference values.

Response: Your suggestion really means a lot to us. In order to better explain our views, according to the reviewer’s suggestion, line 42-43, “Under normal purine diet, the normal SUA level of adults was generally less than 420 μ mol/L7 mg/dl” and line 50-51,And all these indicated risks were independent of normal basal SUA levels. were added.

  1. -Line 57-59 Although drugs such as febuxostat and allopurinol can reduce SUA, they did not decrease the incidence of cardiovascular and cerebrovascular events, and even febuxostat maybe increase these events[12].

A question. If the risks of high uric acid levels are proposed as risk factors for DM2 or its complications, and uricosuric agents do not decrease its incidence, why should we assume that dapagliflozin, due to its uricosuric activity, would?

It was not very clear to me if the dapagliflozin treatment was administered for all 48 weeks of the study or only for 12 weeks.

Response: Thanks for your question. From our perspective, elevated SUA levels can lead to endothelial dysfunction, platelet adhesiveness and production of pro-inflammatory substances, and these are the contributing factors of cardiovascular and cerebrovascular events. On the basis of the widely accepted cardiovascular benefits of dapagliflozin, we speculate that lowering the level of SUA may be one of the ways in which dapagliflozin plays a cardiovascular protective role. We think it is an interesting angle to explore whether febuxostat really increases the risk of cardiovascular and cerebrovascular incidence, or whether there are reasons for resistance of other protection and damage mechanisms ( for instance , long-term use).

Reference:

Kang EH, Kim SC. Cardiovascular Safety of Urate Lowering Therapies. Curr Rheumatol Rep. 2019;21(9):48. doi:10.1007/s11926-019-0843-8.

Thanks for your suggestion, we feel sorry for our inaccurate expression. In fact, the subjects would take dapagliflozin all the time unless there are complications, and we carried out the routine clinic follow up at 12,24 and 48 weeks.

  1. Line-131–133 Abbreviations: GLDs, glucose-lowering drugs; BMI, body mass index; LDL-C, low-density lipo-131 protein cholesterol; HDL-C, high-density lipoprotein cholesterol; HbA1c, glycosylated hemoglo-132 bin; DPP-4i, dipeptidyl peptidase 4 inhibitor; GLP-1RA, glucagon like peptide-1 receptor agonist.

It is suggested to indicate in the graphs the statistical test used and the level of significance. Figure 2 does not indicate whether the paired tests were done after anova, for example.

Response: We think this is an excellent suggestion. Yes, it would be more understandable if we indicate in the graphs the statistical test used and the level of significance.

  1. Line 256–258 In this real-world study, we found that dapagliflozin, as a combination therapy of type 2 diabetes, can reduce the level of SUA and this benefit can be stable for a long time. We suggested that dapagliflozin can be used in combination therapy with T2DM patients with high SUA level.

It seems very clear that the best response to reduce uric acid levels is observed in subjects with high HbA1c values. In how many of the subjects studied in whom dapagliflozin was added relevant reductions were achieved that suggest a decrease in the risks associated with uric acid? How potentially beneficial could the use of dapagliflozin be in prediabetic subjects?

Response: We sincerely appreciate the valuable comments. We did not further explore whether there were decreases in the risks associated with uric acid, which is one of the limitations of our manuscript. We recognize this limitation should be mentioned in the paper, so we added the following sentences “Also, We did not further explore whether there were decreases in the complication risks associated with lowing SUA in T2DM patients, which could need further and more studies.” at line 283-285. Furthermore, Line 275-280, according to relevant literature, Some studies showed that the lowing level of SUA had benefits in patients with T2DM. A randomized, placebo-controlled study from Hong-Kong found that lowering the levels of SUA by allopurinol reduced the rate of eGFR loss. AND Yan Miao suggested that the risk of renal events (a doubling of serum creatinine or end-stage renal disease) was decreased by 6%95%CI 3-10%per 30μmol/L decrement in SUA level in T2DM and nephropathy patients. was added.

We appreciate the reviewers’ insightful suggestion and agree that it would be better to demonstrate that the use of dapagliflozin could have much potential benefits in prediabetic subjects. However, our paper found that for the patients whose SUA levels decreased by 10% or more sub-group, their baseline HbA1c (7.4±0.9 VS 8.3±1.0%; t=-2.610P=0.014) was lower than the sub-group whose SUA levels decreased less than 10%(Line 186-189). In addition, the subjects in our study were not all prediabetic subjects, and such an analysis is beyond the scope of our paper.